# A Loop-Mediated Isothermal Amplification Assay for Rapid Detection of *Pectobacterium aroidearum* that Causes Soft Rot in Konjac

**DOI:** 10.3390/ijms20081937

**Published:** 2019-04-19

**Authors:** Miaomiao Sun, Hao Liu, Junbin Huang, Jinbo Peng, Fuhua Fei, Ya Zhang, Tom Hsiang, Lu Zheng

**Affiliations:** 1The Key Lab of Plant Pathology of Hubei Province, Huazhong Agricultural University, Wuhan 430070, China; miaomiao_01234@163.com (M.S.); hl210@mail.hzau.edu.cn (H.L.); junbinhuang@mail.hzau.edu.cn (J.H.); 2Yichang Academy of Agricultural Science, Yichang 443004, China; pjb98@sina.com (J.P.); 13997689186@163.com (F.F.); 3College of Plant Protection, Hunan Agricultural University, Changsha 410128, China; zhangya230@126.com; 4School of Environmental Sciences, University of Guelph, Guelph, ON N1G 2W1, Canada; thsiang@uoguelph.ca

**Keywords:** comparative genomics, loop-mediated isothermal amplification, konjac, *Pectobacterium aroidearum*, soft-rot bacteria

## Abstract

Bacterial soft rot caused by *Pectobacterium* species is a serious disease in konjac (*Amorphophallus konjac*), a healthy source of starch particularly in East Asia. An effective diagnostic method is crucial to control the disease and reduce losses in konjac production. In this study, we evaluated a loop-mediated isothermal amplification (LAMP) assay with a specific primer set for the rapid and accurate detection of *P. aroidearum*. A comparative genomics approach was used to identify the specific genes suitable for the design of LAMP primers. The candidate target genes were determined through a first-round comparison with a 50-genome nucleotide database, and subjected to a second-round screening with the GenBank NR database. As a result, nine specific genes of *P. aroidearum* were selected for LAMP primer design. After screening of the primers, the primer set 1675-1 was chosen for LAMP detection owing to its high specificity and sensitivity. The LAMP assay could detect the presence of *P. aroidearum* genomic DNA at a concentration as low as 50 fg and 1.2 × 10^4^ CFU/g artificially infected soil within 40 min at 65 °C. Subsequently, this primer set was successfully used to specifically detect *P. aroidearum* in naturally infected and non-symptomatic plant samples or soil samples from the field. This study indicates that a comparative genomic approach may facilitate the development of highly specific primers for LAMP assays, and a LAMP diagnostic assay with the specific primer set 1675-1 should contribute to the rapid and accurate detection of soft-rot disease in konjac at an early stage.

## 1. Introduction

Konjac (*Amorphophallus konjac*) is a perennial herbaceous species that mainly grows in Southeast Asia and Africa [1]. Konjac corm is rich in the hemicellulose glucomannan, which is widely used in many fields, including food additives, chemical engineering, medicine and the oil recovery process. Konjac is also an important source of starch [2,3,4]. China is a major producer of konjac with a planting area of about 1.13 × 10^5^ hectares, particularly in the hilly regions of southern provinces in China [5]. In western Hubei province, bacterial soft rot is the most serious disease of konjac and generally causes a 30%–50% reduction in the corm yield. With continuous culture over several years, the loss of production can even exceed 80% in some plantings [5,6].

As reported previously, bacterial soft rot of konjac was mainly caused by *Pectobacterium carotovorum* subsp. *carotovorum* (Pcc), a member of the Enterobacteriaceae [7]. Pcc has tended to serve as a catchall for pectobacterial isolates differing from the specific descriptions of the other pectobacterial taxa [8]. There are many examples of misidentification in the Pcc literature. For example, the genome sequenced strain PC1 is actually *Pectobacterium aroidearum*, but misidentified as Pcc [9]. Bacterial strain M8, isolated from konjac soft rot samples in Shaanxi province, was clustered together with *P. aroidearum* strain PC1 using phylogenetic analysis [10]. In this study, we confirmed that the main pathogen causing bacterial soft rot on konjac from Yichang, China is actually *P. aroidearum* using multilocus sequence typing (MLST) analysis (Appendix A). *Pectobacterium* usually exists in soils with a broad range of hosts [11]. Once the konjac corms are infected, there is no effective method to control bacterial soft rot. A common practice is to use fungicide-treated, healthy corms before planting. Implementation of phytosanitary control measures can be drastic and relies upon accurate detection and diagnosis, preferably with quantitative measurement of pathogen inoculum presence. Rapid and accurate detection of the soft-rot pathogen in the field in the early stage can provide researchers and growers with reliable information on the infection status of their fields, and hence allow for timely control measures.

With advances in molecular biology techniques, conventional polymerase chain reaction (PCR), multiplex PCR and real-time PCR have been used to achieve a more accurate diagnosis of *Pectobacterium* species [12,13,14]. Previous studies mostly attempted to distinguish the soft-rot pathogens using the genes associated with virulence, such as *pel* [15], or *cfa6* [16], taxon-specific regions, such as 16S rRNA or 16S-23S intergenic sequences [7,17], or the housekeeping gene *rhsA* [18]. However, to date, there have been no published studies on the specific detection of *P. aroidearum*.

Application of genomic-based approaches for mining specific detection targets has been reported in several phytopathogens, including *Pseudomonas fuscovaginae* [19], *Xanthomonas arboricola* pv. *pruni* [20], *Calonectria henricotiae*, *C. pseudonaviculata* [21] and *Ustilaginoidea virens* [22]. In this study, we developed a loop-mediated isothermal amplification (LAMP) method for the specific detection of *P. aroidearum* based on primers derived from comparative genomics. The LAMP assay requires a *Bst* DNA polymerase with strand displacement activity and a set of four to six primers specifically designed to recognize six regions on the target DNA to amplify DNA with high specificity, sensitivity and efficiency under isothermal conditions at 60–65 °C in less than 1 h [23,24,25,26]. The amplification results of the LAMP assay can be directly observed by eye, since positive amplification produces a large amount of by-product pyrophosphate ions, which can be combined with magnesium ions in the solution to form a white precipitate of magnesium pyrophosphate, while negative reactions result in no white precipitate [27]. Moreover, when SYBR Green I is added to the products, the color of positive amplification products turns fluorescent green while the negative control remains orange [26]. In addition, LAMP analysis can be done within an hour given a stable heat source at 60–65 °C, such as a thermos, and it does not require other specialized equipment. In contrast, PCR would require an energy source for the thermocycler and would take longer. Therefore, because of the specificity, sensitivity, efficiency and simplicity of the LAMP assay, it is more suitable for direct application in the field compared with PCR-based methods.

To date, LAMP assays have been applied to the diagnosis of some *Pectobacterium* species [28]. However, these assays are not capable of specifically detecting *P. aroidearum*. In this study, we developed a novel gene target specific to *P. aroidearum* for LAMP detection using a comparative genomics approach. The specificity was confirmed through testing of various *Pectobacterium* species or subspecies, other pathogenic bacteria and konjac rhizosphere bacteria. Finally, we validated the applicability of the LAMP assay in naturally infected konjac and soil samples from the field.

## 2. Results

### 2.1. Isolation and Pathogenicity of Konjac Soft Rot Pathogen

From 2017 to 2018, we obtained 115 isolations from konjac soft rot samples collected from Yichang and Enshi, in western Hubei province. There were 35 strains sharing a morphotype that was assumed to be *Pectobacterium*, 15 strains resembling *Dickeya*, and the other 65 strains were assumed to be *Bacillus* and other bacteria based on gross morphological features. All 115 strains were inoculated onto konjac corms. The results showed that 50 strains caused rot 24 h after inoculation, and the normally white tissue turned yellowish brown or grayish white, showing a sticky paste texture and a distasteful odor, which was consistent with the field symptoms (data not shown). After morphological observation, the 50 strains were divided into two types, one of which had 35 strains, with colonies on LB solid medium that were milky white, nearly round, with a smooth surface and neat edges, and slightly raised in the center of the colonies. The other 15 strains had colonies on LB solid medium that were milky white and wavy edges. We first used 16S rDNA sequencing and the 35 strains were placed as *Pectobacterium*, while the 15 strains were placed as *Dickeya* (*D. fangzhongdai*). MLST analysis was then done to reveal that the 35 strains were *P. aroidearum* (Appendix A). We also tested the pathogenicity of Pcc strain (CGMCC1.141) on konjac, and found that it was unable to cause disease on konjac. Thus, *P. aroidearum* is the main soft rot pathogen of konjac in western Hubei, province, accounting for 70%, while *Dickeya* accounts for 30%. Therefore, the following experiments used *P. aroidearum* as the subject for further study.

### 2.2. Mining of Specific Genes in the P. aroidearum Genome for LAMP Primer Design

The genome of *P. aroidearum* strain PC1 (GenBank accession CP001657.1) was used for mining specific genes. In the first-round screening, 4393 gene sequences of *P. aroidearum* were used as queries against a 50-genome nucleotide database using Standalone TBLASTX. The results yielded 132 genes that had no matches in the database. These unmatched genes were then used as translated nucleotide queries against the GenBank NR protein database using Standalone BLASTX for the second-round screening. Among these, 30 genes had no matches in the NR protein database, and these were thus suitable for designing and testing of specific primers. Finally, nine *P. aroidearum* genes were chosen as candidate targets for designing specific primers (Figure 1, Table 1).

To briefly evaluate the specificity of the primers, all designed LAMP primer sets were first evaluated by conventional PCR amplification using the DNA of *Pectobacterium* spp. (*P. aroidearum,* Pcc, *P. carotovorum* subsp. *actinidiae* and *P. carotovorum* subsp. *brasiliense*), *Dickeya* spp. (*D. fangzhongdai*, *D. chrysanthemi* and *D. zeae*), and *Erwinia* spp. (*E. amylovora* and *E. pyrifoliae*) with outer primers F3 and B3. After screening, a total of six primer sets were specific to *P. aroidearum* by conventional PCR. Then, the primers in all these LAMP primer sets were evaluated in LAMP assay. Finally, the primer set 1675-1 from the candidate target PC1_1675 (Table 2) was selected since it showed high specificity without false positive amplifications, and the positive amplifications consistently showed bright and ladder-like bands in agarose gel electrophoresis. To further validate the specificity of LAMP products, the products were sequenced and the alignment results showed that the sequence was exactly a portion of gene PC1_1675 of *P. aroidearum*. Therefore, primer set 1675-1 was chosen for the detection of soft-rot bacteria *P. aroidearum* in konjac.

### 2.3. Specificity of LAMP Assay

To assess the specificity of primer set 1675-1 for *P. aroidearum*, we conducted LAMP assays with the primer set to test the DNA from 24 *P. aroidearum* strains, three strains of *P. carotovorum* subspecies (Pcc, *P. carotovorum* subsp. actinidiae and *P. carotovorum* subsp. brasiliense), 17 strains of *Dickeya* spp. (*D. fangzhongdai*, *D. chrysanthemi* and *D. zeae*), two strains of *Erwinia* spp. (*Erwinia amylovora* and *E. pyrifoliae*), and another three plant-associated bacterial pathogens (*Ralstonia solanacearum*, *Acidovorax citrulli* and *Xanthomonas oryzae* pv. oryzae), as well as 21 konjac rhizosphere bacteria in the environmental samples (Table 3). The results showed that all *P. aroidearum* strains had positive amplifications with bright green fluorescence when SYBR Green I was added, while other species had no amplifications and the products remained light orange after the addition of SYBR Green I (Table 3). Furthermore, when the outer primers (F3 and B3) of primer set 1675-1 were used for conventional PCR to assess the specificity, only *P. aroidearum* strains showed a 218-bp specific band, while other strains showed no amplifications (Figure 2). These results suggested that the primer set 1675-1 could be used for the specific detection of *P. aroidearum*.

### 2.4. Sensitivity of LAMP Assay

To evaluate the sensitivity of the LAMP assay, genomic DNA was determined in 10-fold serial dilutions (from 50 ng/μL to 5 fg/μL) of *P. aroidearum* strain YCC 1. The results indicated that the detection limit of LAMP assay was 50 fg; whereas for conventional PCR, when the DNA was diluted to 5 pg, there was only a weak amplification band (Figure 2). Therefore, the sensitivity of LAMP detection was 100-fold greater than conventional PCR.

### 2.5. LAMP Detection of Artificially Inoculated Soil Samples

The artificially inoculated soil samples containing different amounts of fresh bacterial suspension of *P. aroidearum* were prepared, and then the DNA was extracted and evaluated by LAMP assay and conventional PCR, respectively (Figure 3). The LAMP assay could detect *P. aroidearum* at a concentration as low as 1.2 × 10^4^ CFU/g from artificially inoculated soil samples; in contrast, conventional PCR could only detect a concentration of 1.2 × 10^5^ CFU/g. As for controls, no *P. aroidearum* DNA was detected in the uninoculated soil samples by either the LAMP assay or conventional PCR. Hence, the sensitivity of LAMP detection was 10 times that of conventional PCR for artificially inoculated soil samples.

### 2.6. Application of LAMP to Konjac Plants and Rhizosphere Soil in the Field

To demonstrate the applicability of LAMP in the field, konjac corm and soil samples were collected from Wufeng county, Yichang, Hubei province, China in June 2018. For all five konjac corm samples with visible soft rot symptoms, positive amplification products were obtained with primer set 1675-1 in both LAMP assay and conventional PCR. Among the five konjac corm samples without visible soft rot symptoms, positive amplification products were obtained for four samples in a LAMP assay whereas for only one sample in conventional PCR detection (Figure 4).

Soil samples were also assessed using LAMP assay and conventional PCR. Among 30 soil samples, positive amplification products were obtained for 13 samples (43.3%) in LAMP assay and for nine samples (30%) in conventional PCR detection using the primer set 1675-1 (Figure 5). The results indicated that the LAMP assay using the designed primer set in this study has a higher sensitivity in detecting *P. aroidearum* than conventional PCR.

## 3. Discussion

In this study, we evaluated a LAMP assay for the detection of konjac soft rot pathogen, *P. aroidearum* using a novel target that we identified by applying a comparative genomics approach. Comparison of complete genomes provides an efficient way for uncovering species-specific genomic regions. Comparison programs, such as BLAST, a frequently used tool for searching sequence similarity, have been applied to mine specific regions for the detection of bacterial and fungal pathogens [19,20,21,22,29,30,31]. Compared with previous studies which used comparative genomics methods to mine the specific targets of bacteria [19,20], we performed two rounds of screening to select specific regions. The initial round of BLAST resulted in a small number of relatively specific targets, excluding the majority of *P. aroidearum* genes. These potentially unique genes were then screened against the huge GenBank NR database for the second round to further narrow down the gene candidates with higher sensitivity and specificity. This two-tiered approach can save time and reduce computational complexity in contrast to the direct comparison of complete gene sets against the GenBank database.

Previous studies have reported on the LAMP-based detection of *P. carotovorum* [28]. However, detection technology regarding *P. aroidearum* has not been previously reported. Thus, this is the first description of specific detection of this konjac soft rot pathogen, and we evaluated a LAMP detection assay specific for *P. aroidearum* which can detect *P. aroidearum* infections of konjac at an early stage of the disease. Compared with conventional PCR and real-time qPCR, the LAMP assay in this study can achieve rapid DNA amplifications under isothermal conditions, and does not require expensive and elaborate laboratory equipment, or highly skilled operations [32]. Therefore, the LAMP assay is more suitable for on-site detection in the field than the other methods.

After screening, primer set 1675-1 of *P. aroidearum* was selected because of its high repeatability, efficiency and specificity. The target gene sequence PC1_1675, from which primer set 1675-1 was derived, was a glycosyl transferase gene found in *P. aroidearum*, but not in any other sequence available on GenBank. The glycosyl transferases are present in all living organisms, but they show low sequence identity [33]. In addition, loop primers LF and LB were designed to accelerate the LAMP reaction and improve the detection efficiency [25]. The reaction time was significantly shortened from 60 min to 30 min. LAMP detection with the primer set showed a higher sensitivity than conventional PCR. The detection limit of the LAMP method developed in this study was 50 fg/μL of *P. aroidearum* genomic DNA, which was 100 times more sensitive that of the conventional PCR. In addition, the LAMP assay could detect *P. aroidearum* at a concentration as low as 1.2 × 10^4^ CFU/g of bacterial cells in artificially inoculated soil samples, which is 10 times more sensitive that of conventional PCR. Because of its high sensitivity, LAMP assay has been widely used for the early detection of latent infections by agricultural pathogens [34]. We also applied LAMP assay for the detection of *P. aroidearum* in konjac and soil samples in the field. For all konjac corms with visible symptoms, positive amplification results were obtained using primer set 1675-1, indicating the applicability of LAMP assay in the field. Under normal conditions, pathogenic bacteria have certain latent periods and do not directly cause symptoms in the host until they reach a certain density and encounter favorable conditions for disease development. Therefore, the rapid LAMP detection of non-symptomatic latent infections provides critical information for early prevention and control of disease outbreaks. Further work is needed to confirm the relationships between environment conditions and disease severity after latent infection detected.

In conclusion, we developed a primer set with high specificity in the LAMP assay to detect the konjac soft rot pathogen *P. aroidearum* using a comparative genomics approach. This study also revealed that such approaches may contribute to the development of highly specific primers for LAMP assays. The LAMP primer set 1675-1 showed high sensitivity and specificity for *P. aroidearum* and thus can be further used for early diagnosis of the pathogen in symptomless plants and also for evaluation of soft rot resistance in konjac cultivars.

## 4. Materials and Methods

### 4.1. Isolation and Pathogenicity of Konjac Soft Rot Pathogen

From August 2017 to September 2018, konjac soft rot samples were collected from Yichang and Enshi, the main konjac production areas in western Hubei province. Potential pathogens were isolated and purified by growth on LB solid medium. After incubation at 28 °C for 24 h, single colonies were suspended in 4 mL LB liquid medium and incubated at 28 °C for 12 h. Then 20 μL of this 1 × 10^8^ CFU/mL suspension were placed onto healthy washed konjac corms, and symptoms were evaluated 24 and 48 h later. For strains that caused a soft rot of konjac corms in our pathogenicity test, 16S rDNA sequencing and MLST analysis were used to identify these strains.

### 4.2. Bacterial Strains and DNA Preparation

Bacterial strains used in this study included 24 strains of *P. aroidearum*, 15 strains of *D. fangzhongdai*, 21 strains of konjac rhizosphere bacteria in the environmental samples and 10 other bacterial pathogens of plants (Table 3).

The bacterial strains were grown on Luria-Bertani (LB) medium (10 g/L tryptone, 10 g/L NaCl, 5 g/L yeast extract and 18 g/L agar), and incubated at 28 °C for 24 h. Single colonies were then picked and suspended in liquid LB medium at 28 °C for 24 h. Bacterial gDNA was extracted using the TIANamp Bacteria DNA Kit (TIANGEN, Beijing, China) following the manufacturer’s instructions. DNA extracts were quantified at 260 and 280 nm using a NanoVue Plus spectrophotometer (GE Healthcare Life Sciences, Piscataway, NJ, USA). Extracted DNA was stored at −20°C until use as a template for LAMP or PCR reaction.

### 4.3. Soil Samples and DNA Preparation

The fresh *P. aroidearum* bacterial suspension was diluted to different concentrations (6 × 10^8^, 6 × 10^7^, 6 × 10^6^, 6 × 10^5^, 6 × 10^4^, 6 × 10^3^ and 6 × 10^2^ CFU/mL) using a turbidity counting method. Each 100 µL bacterial suspension was placed onto 0.5 g of sterilized soil in a 10-mL centrifuge tube. The soil treated with LB broth was used as a blank control. Soil DNA was extracted using the EZNA^®^ Soil DNA kit (Omega Bio-Tek, Norcross, GA, USA).

### 4.4. Sources of Genome Sequence Data

The genome of *P. aroidearum* strain PC1 (GenBank accession CP001657.1) is publicly available. To identify the specific genes of *P. aroidearum*, we set up a 50-genome database for the first round screening. These 50 genomes included *Pectobacterium* spp. and its associated subspecies (*P. carotovorum* subsp. *brasiliensis* and *P. carotovorum* subsp. *actinidiae*), *Dickeya* spp. (*D. solani* and *D. zeae*), *Erwinia* spp. (*E. amylovora*, *E. tracheiphila*, and *E. persicina*), other pathogenic bacteria (*Agrobacterium tumefaciens*, *Pantoea ananatis*, *P. stewartii* subsp. *stewartii*, and *Ralstonia solanacearum*), and soil bacteria (*Bacillus subtilis*) (Appendix A). All genomes used in this study were downloaded from NCBI (https://www.ncbi.nlm.nih.gov/genome/) and JGI (https://genome.jgi.doe.gov/portal/) in July 2017. The GenBank NR database used for the second round screening was also downloaded from NCBI (ftp://ftp.ncbi.nlm.nih.gov/blast/db/) in August 2017.

### 4.5. Mining of Specific Targets of P. Aroidearum

Standalone BLAST (version 2.6.0+) was set up on a Microsoft Windows PC system, and a BLAST alignment database was generated for each genome and the associated predicted gene database. The 4393 gene sequences of *P. aroidearum* were searched against the 50-genome nucleotide database using Standalone TBLASTX for the first round of screening. The e-value of 1e-5 was set as the threshold for homology. PERL scripts were used to parse the BLAST output files [35]. The genes having no matches with the 50-genome database were chosen for the second round of screening against the GenBank NR database using Standalone BLASTX. After the second round of screening, the genes satisfying the following conditions were selected: (i) With lengths ranging from 200 bp to 2000 bp; (ii) present in the *P. aroidearum* strain PC1, but not Pcc strain genomes in the NR database. These genes were considered as specific and chosen as candidate targets for designing LAMP primers (Table 1).

### 4.6. Design and Screening of LAMP Primers

To select the optimal LAMP primers, nine candidate genes for *P. aroidearum* were selected, and the LAMP primers used in this study were designed with PrimerExplorer V5 (http://primerexplorer.jp/e/; Eiken Chemical Co., Ltd., Tokyo, Japan). The LAMP primer sets each included four to six primers (outer primers F3 and B3, inner primers FIP and BIP, and loop primers LF and LB). The primers were synthesized by TIANYI HUIYUAN (Wuhan, China).

### 4.7. LAMP Reaction and Product Validation

LAMP reaction mixture contained 1.6 µM each of FIP and BIP, 0.4 µM each of LF and LB, 0.2 µM each of F3 and B3, 1.4 mM each of dNTPs (TaKaRa, Dalian, China), 1 M betaine (Sigma, St. Louis, MO, USA), 1× isothermal amplification buffer (20 mM Tris-HCl, 10 mM (NH_4_)_2_SO_4_, 50 mM KCl, 2 mM MgSO_4_, 0.1% Tween-20, pH 8.8), 6 mM MgSO_4_, 8 U of Bst 2.0 WarmStart polymerase (New England Biolabs, Beverly, MA, USA), 1 µL of template DNA, and sterile deionized water to a final volume of 25 μL. For the negative control, sterilized deionized water was used to replace the DNA template. These reaction mixtures were respectively put into 200 μL centrifuge tubes, which were placed in a T100 Thermal Cycler (Bio-Rad Laboratories Inc., Hercules, CA, USA) at a constant temperature of 65 °C for 40 min. Then, the Bst 2.0 WarmStart polymerase was heat inactivated at 80 °C for 5 min to stop the reactions. When the reaction products were cooled, 1 µL of 1/10 diluted original SYBR Green I (Solarbio, Beijing, China) was added into every tube for color reaction.

To verify whether the LAMP product was the correct target fragment, conventional PCR amplification was performed with the outer primers F3 and B3. The 218-bp fragment of *P. aroidearum* was amplified, and was cloned into pMD18-T (Takara, Kyoto, Japan) and sequenced.

### 4.8. Conventional PCR Assay

Conventional PCR was performed with the outer primer pair F3 and B3 of *P. aroidearum*. The PCR mixture contained 12.5 μL of 2×Hieff^TM^ PCR Master Mix (YEASEN, Shanghai, China), 0.4 μM of each primer, 1 μL of DNA template, and sterile deionized water to a final volume of 25 μL. The mixtures were amplified in a T100 Thermal Cycler with the following conditions: Initial denaturation at 94 °C for 5 min, followed by 30 cycles of 94 °C for 30 s, 55 °C for 30 s, and 72 °C for 30 s, and 72 °C for 5 min as a final extension step. The PCR products were visualized after electrophoresis in 2% agarose gels.

### 4.9. LAMP Specificity and Sensitivity Assays

To confirm whether the primers were specific to *P. aroidearum*, we evaluated a total of 70 representative strains, including strains of *Pectobacterium* spp., *Dickeya* spp., *Erwinia* spp., several plant-associated pathogens, and konjac rhizosphere bacteria in the environmental samples (Table 3). Primers of 16S ribosomal DNA (rDNA) (27f/1492r) were amplified to first test the quality of the DNA. To assess the sensitivity of the LAMP assay, purified genomic DNA of *P. aroidearum* strain YCC 1 was 10-fold serially diluted from 50 ng/μL to 5 fg/μL. In addition, DNA from artificially inoculated soil samples was tested for LAMP sensitivity. The results of LAMP assay were visualized by the SYBR Green I staining. Furthermore, the 10-fold serially diluted DNA was amplified by conventional PCR for a comparison of sensitivity with LAMP assay. All samples were tested at least twice for confirmation.

### 4.10. Application of LAMP Assay in Naturally Infected Konjac and Field Rhizosphere Soil

To evaluate the application of LAMP assay in naturally infested konjac corms and soil in the field, 10 konjac corms, including five samples with visible symptoms and another five samples without visible symptoms, and 30 soil samples were randomly collected from three fields in Yichang city, Hubei province, China in June 2018.

For each corm sample, a small piece of konjac corm was transferred into a 2 mL tube, and 1 mL of TE buffer was added with brief vortexing, followed by incubation at 95 °C for 15 min. Finally, 1 μL of the solution was used as the DNA template for the LAMP or conventional PCR assay. For the soil samples, the DNA was extracted using the EZNA^®^ Soil DNA kit for both LAMP assay and conventional PCR amplification using primer set for *P. aroidearum*. All samples were tested at least twice.

## Figures and Tables

**Figure 1 ijms-20-01937-f001:**
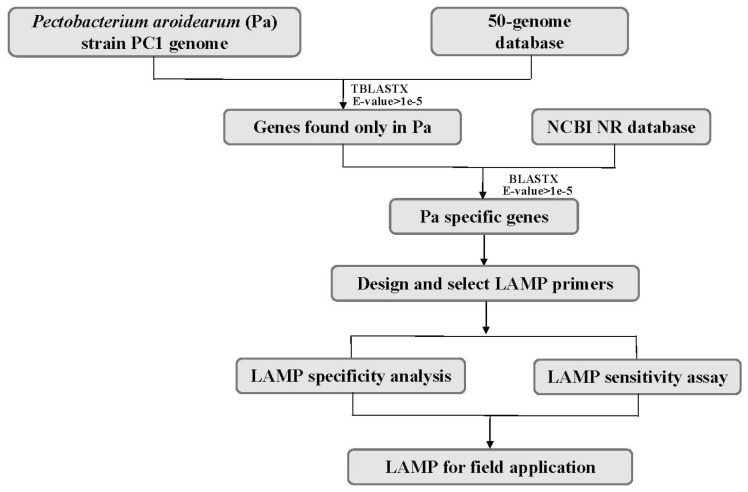
Flowchart for mining specific genes of *P. aroidearum* using a comparative genomic approach.

**Figure 2 ijms-20-01937-f002:**
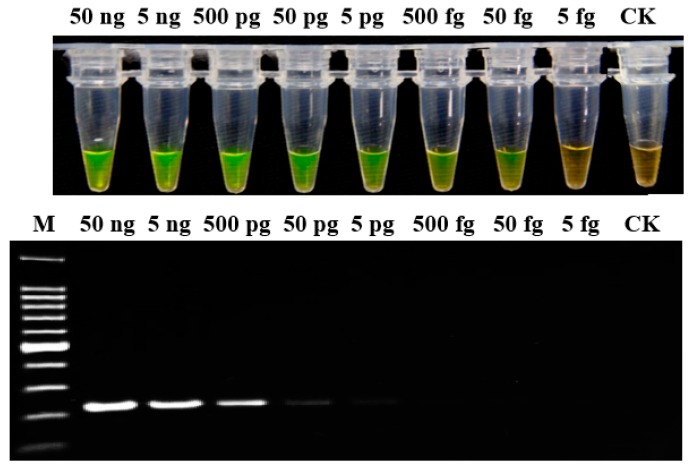
Sensitivity analysis of LAMP and conventional PCR assays for genomic DNA of *P. aroidearum* using primer set 1675-1. Genomic DNA was determined in 10-fold serial dilutions, from 50 ng/μL to 5 fg/μL. CK: Negative control. M: 100 bp DNA Ladder (TSINGKE, Wuhan, China).

**Figure 3 ijms-20-01937-f003:**
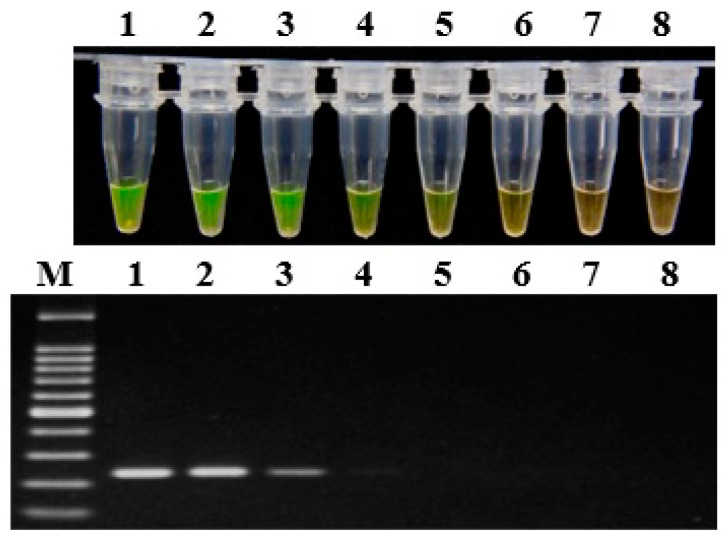
Sensitivity of LAMP (top) and conventional PCR assays (bottom) on artificially infected soil with *P. aroidearum* using primer set 1675-1. Lane 1–7: Serial 10-fold dilutions ranging from 1.2 × 10^8^ to 1.2 × 10^2^ CFU/g of artificially infested soil. Lane 8: Negative control. M: 100 bp DNA Ladder (TSINGKE, Wuhan, China).

**Figure 4 ijms-20-01937-f004:**
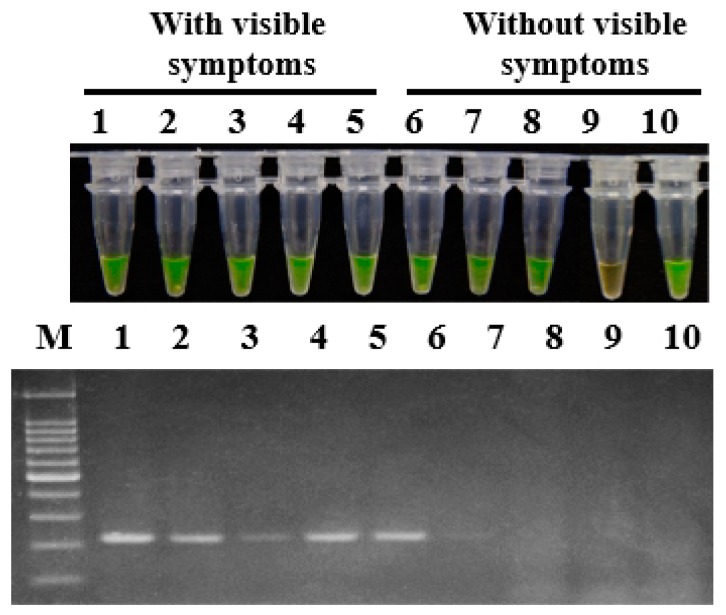
Application of LAMP and conventional PCR on naturally infected konjac corms. The konjac corm samples were collected from Yichang city, Hubei province in June 2018. The primer set 1675-1 was used in the assays for detection of *P. aroidearum*. Lane 1–5: Konjac corms with visible symptoms. Lane 6–10: Konjac corms without visible symptoms. M: 100 bp DNA Ladder (TSINGKE, Wuhan, China).

**Figure 5 ijms-20-01937-f005:**
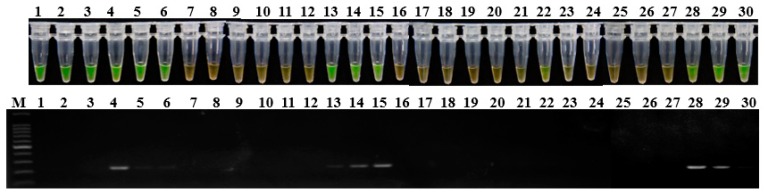
Application of LAMP (top) and conventional PCR assays (bottom) in soil from konjac fields. The rhizosphere soil samples were randomly collected from three fields in Yichang city, Hubei province in June 2018, and assayed for *P. aroidearum* using primer set 1675-1. Lane 1–30: Rhizosphere soil samples. M: 100 bp DNA Ladder (TSINGKE, Wuhan, China).

**Table 1 ijms-20-01937-t001:** Summary of genomic regions of *Pectobacterium aroidearum* used to design loop-mediated isothermal amplification (LAMP) primers in this study.

Gene No.	Length (bp)	BLASTX Prediction	Predicted Domain ^a^	Signal Peptide ^b^
PC1_0970	303	Hypothetical protein	YCII	No
PC1_1421	1320	Hypothetical protein	No	Yes
PC1_1622	452	Hypothetical protein	No	No
PC1_1675	1809	Glycosyl transferase	Glyco_transf_8	No
PC1_2056	891	Hypothetical protein	No	Yes
PC1_2248	792	Hypothetical protein	No	No
PC1_2623	831	Hypothetical protein	DUF4393	No
PC1_2625	345	Hypothetical protein	No	No
PC1_2831	303	Hypothetical protein	No	Yes

^a^ Protein domain was predicted by SMART (http://smart.embl-heidelberg.de/). ^b^ Signal peptide was predicted by Signal P (http://www.cbs.dtu.dk/services/SignalP/).

**Table 2 ijms-20-01937-t002:** LAMP primer set 1675-1 used in this study.

Primer	Sequence (5′-3′)	Description
F3	GCACAAGCTTGACTGCATAC	Outer primer
B3	TGGCGAGTTGTCCCCATAG	Outer primer
FIP	CGCCGTATCGGCACAGAAGAAAGCTTGGCGTTTCTCTCTCA	Inner primer
BIP	GCCCTCATCTCGCTGGCAATCATGTGCTTCCGGCAACAC	Inner primer
LF	AGAAACGGGATGGGGTGG	Loop primer
LB	GCCATCGAGCGTAGCGAA	Loop primer

**Table 3 ijms-20-01937-t003:** Bacterial strains used for the specificity detection of LAMP primer set 1675-1 for *P. aroidearum*.

Species	Strain	Host	Location	LAMP ^a^ (1675-1)	PCR (1675F3/B3)
*P. aroidearum*	YCC1	Konjac	Yichang, Hubei	+	+
*P. aroidearum*	YCC2	Konjac	Yichang, Hubei	+	+
*P. aroidearum*	YCC3	Konjac	Yichang, Hubei	+	+
*P. aroidearum*	YCC4	Konjac	Yichang, Hubei	+	+
*P. aroidearum*	YCC5	Konjac	Yichang, Hubei	+	+
*P. aroidearum*	YCC6	Konjac	Yichang, Hubei	+	+
*P. aroidearum*	YCC7	Konjac	Yichang, Hubei	+	+
*P. aroidearum*	YCC8	Konjac	Yichang, Hubei	+	+
*P. aroidearum*	YCC9	Konjac	Yichang, Hubei	+	+
*P. aroidearum*	YCC10	Konjac	Yichang, Hubei	+	+
*P. aroidearum*	YCC11	Konjac	Yichang, Hubei	+	+
*P. aroidearum*	YCC12	Konjac	Yichang, Hubei	+	+
*P. aroidearum*	YCC13	Konjac	Yichang, Hubei	+	+
*P. aroidearum*	YCC14	Konjac	Enshi, Hubei	+	+
*P. aroidearum*	NS-1	Konjac	Enshi, Hubei	+	+
*P. aroidearum*	NS-2	Konjac	Enshi, Hubei	+	+
*P. aroidearum*	NS-3	Konjac	Enshi, Hubei	+	+
*P. aroidearum*	NS-4	Konjac	Enshi, Hubei	+	+
*P. aroidearum*	NS-5	Konjac	Enshi, Hubei	+	+
*P. aroidearum*	WF-1	Konjac	Yichang, Hubei	+	+
*P. aroidearum*	WF-2	Konjac	Yichang, Hubei	+	+
*P. aroidearum*	WF-3	Konjac	Yichang, Hubei	+	+
*P. aroidearum*	WF-4	Konjac	Yichang, Hubei	+	+
*P. aroidearum*	WF-5	Konjac	Yichang, Hubei	+	+
*D. fangzhongdai*	YCH1	Konjac	Yichang, Hubei	−	−
*D. fangzhongdai*	YCH2	Konjac	Yichang, Hubei	−	−
*D. fangzhongdai*	YCH3	Konjac	Yichang, Hubei	−	−
*D. fangzhongdai*	YCH4	Konjac	Yichang, Hubei	−	−
*D. fangzhongdai*	YCH5	Konjac	Yichang, Hubei	−	−
*D. fangzhongdai*	YCH6	Konjac	Yichang, Hubei	−	−
*D. fangzhongdai*	YCH7	Konjac	Yichang, Hubei	−	−
*D. fangzhongdai*	YCH8	Konjac	Yichang, Hubei	−	−
*D. fangzhongdai*	YCH9	Konjac	Yichang, Hubei	−	−
*D. fangzhongdai*	WFH-1	Konjac	Yichang, Hubei	−	−
*D. fangzhongdai*	WFH-2	Konjac	Yichang, Hubei	−	−
*D. fangzhongdai*	WFH-3	Konjac	Yichang, Hubei	−	−
*D. fangzhongdai*	WFH-4	Konjac	Yichang, Hubei	−	−
*D. fangzhongdai*	WFH-5	Konjac	Yichang, Hubei	−	−
*D. fangzhongdai*	WFH-6	Konjac	Yichang, Hubei	−	−
Pcc	CGMCC1.141	Cabbage	Beijing	−	−
*P. carotovorum* subsp. *actinidiae*	HN-1	Chili	Hunan	−	−
*P. carotovorum* subsp. *brasiliense*	YCB1	Konjac	Yichang, Hubei	−	−
*D. chrysanthemi*	CGMCC1.7280	Unknown	Beijing	−	−
*D. zeae*	CGMCC1.3614	Unknown	Beijing	−	−
*Erwinia amylovora*	CGMCC1.7276	Unknown	Beijing	−	−
*E. pyrifoliae*	CGMCC1.7277	Unknown	Beijing	−	−
*Ralstonia solanacearum*	GMI1000	Potato	Wuhan, Hubei	−	−
*Acidovorax citrulli*	HN-2	Watermelon	Hunan	−	−
*Xanthomonas oryzae* pv. *oryzae*	HW-67	Rice	Wuhan, Hubei	−	−
*Kosakonia pseudosacchari*	M1	Konjac	Yichang, Hubei	−	−
*Acinetobacter calcoaceticus*	M2	Konjac	Yichang, Hubei	−	−
*Bacillus aryabhattai*	M3	Konjac	Yichang, Hubei	−	−
*B. firmus*	M4	Konjac	Yichang, Hubei	−	−
*B. simplex*	M5	Konjac	Yichang, Hubei	−	−
*Citrobacter farmeri*	M6	Konjac	Yichang, Hubei	−	−
*C. freundii*	M7	Konjac	Yichang, Hubei	−	−
*Enterobacter aerogenes*	M8	Konjac	Yichang, Hubei	−	−
*E. asburiae*	M9	Konjac	Yichang, Hubei	−	−
*Enterobacter* sp.	M10	Konjac	Yichang, Hubei	−	−
*Klebsiella oxytoca*	M11	Konjac	Yichang, Hubei	−	−
*K. pneumoniae*	M12	Konjac	Yichang, Hubei	−	−
*Morganella morganii*	M13	Konjac	Yichang, Hubei	−	−
*Paenibacillus polymyxa*	M14	Konjac	Yichang, Hubei	−	−
*Pantoea agglomerans*	M15	Konjac	Yichang, Hubei	−	−
*Providencia rettgeri*	M16	Konjac	Yichang, Hubei	−	−
*Pseudomonas plecoglossicida*	M17	Konjac	Yichang, Hubei	−	−
*P. putida*	M18	Konjac	Yichang, Hubei	−	−
*Raoultella ornithinolytica*	M19	Konjac	Yichang, Hubei	−	−
*R. planticola*	M20	Konjac	Yichang, Hubei	−	−
*R. terrigena*	M21	Konjac	Yichang, Hubei	−	−

^a^ +: positive reaction; −: negative reaction.

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
