# Peer review of "A Loop-Mediated Isothermal Amplification Assay for Rapid Detection of Pectobacterium aroidearum that Causes Soft Rot in Konjac"

_ijms, 2019, doi:10.3390/ijms20081937_

Round 1
Reviewer 1 Report
The comments are on the PDF file.

Author Response
Reviewer #1
All comments were made as track changes in a revision of the manuscript.
Comment 1: L49-51: Revise this sentence to make the statement clear. "In western Hubei province, as the most serious disease of konjac, bacterial soft rot generally causes a 30%–50% reduction in the corm yield of konjac, continuous cropping fields, and the loss of production even exceeds 80% in some planting areas [5]. [6]."
Response: The sentence has been revised as "In western Hubei province, bacterial soft rot is the most serious disease of konjac and generally causes a 30%–50% reduction in the corm yield. With continuous culture over several years, the loss of production can even exceed 80% in some plantings [5][6]."
Comment 2: L88: Add a sentence or two to highlight why LAMP can be applicable in field, but not PCR.
Response: As suggested, we added some sentences to highlight why LAMP can be applicable in field as follows: “In addition, LAMP analysis can be done within an hour given a stable heat source at 60-65 °C such as a thermos, and it does not require other specialized equipment. In contrast, PCR would require an energy source for the thermocycler and would take longer. Therefore, because of specificity, sensitivity, efficiency and simplicity of the LAMP assay, it is more suitable for direct application in the field compared with PCR-based methods.”
Comment 3: L91: Why previous studies are not capable of detecting P. aroidearum species?
Response: In the revised manuscript , we wrote on Page 2, Line 69-72: "Previous studies mostly attempted to distinguish the soft-rot pathogens using the genes associated with virulence such as pel [13], or cfa6 14], taxon-specific regions such as 16S rRNA or 16S-23S intergenic sequences [7,16], or the housekeeping gene rhsA [17]. However, to date, there have been no published studies on specific detection of P. aroidearum".
Since these sequences could not resolve P. aroidearum from related taxa. Nabhan et al. (2013) used eight housekeeping genes (acnA, icd, gapA, mdh, mtlD, pgi, proA and rpoS) to name this new species and make it separate from other related species. In this study, we decided to uncover genes that could specifically distinguish P. aroidearum from other taxa using a comparative genomics gene-mining method. As we state in the manuscript, we found 30 genes that were unique to P. aroidearum, and continued to test 9 of them and selected 1 for use in LAMP analysis.
Comment 4: L98-99: Why this specific strain was chosen?
Response: Among P. aroidearum isolates, only the genome of isolate PC1 was available, and hence this was reason for choosing it.
Comment 5: Page 8: The figure on Page 8 seems incomplete and figure title is missing.
Response: The figure on Page 8 was deleted in the previous revision (as per previous reviewer suggestions), but the add-remove markings did not clearly show this deletion.
Comment 6: Figure 2: If possible please add the concentrations of each tube over the numbers on the figure.
Response: As suggested, we have generated a figure with the concentrations above the numbers. Please see revised Figure 2 in the manuscript.
Comment 7: L300: More discussion is required about the selected target fragment.
Response: We have added the following information to the Discussion, Page 9 Line 237: "After screening, primer set 1704-1 of P. aroidearum was selected because of its high repeatability, efficiency and specificity. The target gene sequence PC1_1675, from which primer set 1704-1 was derived, was a glycosyl transferase gene found in P. aroidearum but not in any other sequence available on GenBank. The glycosyl transferases are present in all living organisms, but they show low sequence identity [33].
Reviewer 2 Report
The manuscript by Sun and colleagues is a revised manuscript (based on the multiple corrections) that describes the development of a sensitive and specific LAMP test for Pectobacterium aroidearum, a cause of soft rot in the konjac plant. Overall the manuscript is clearly written but there are apparent errors in locus ID (for the critical target identified and exploited) and the relationship of Pa versus Pcc needs to be better explained—especially since the manuscript has instances where Pa is being described as Pcc (mainly in the methods).
Specific Points (Scientific)
1. Throughout the ms (mostly), the Pcc language has been replaced by P. aroidearum. As originally written, the manuscript appears to have developed this assay for Pcc, but this has now been revised. Although there appears to be considerable misidentification in the Pcc group, the authors need to provide further explanation about whether bona fide Pcc isolates cause soft rot in konjac (as in ref 7) or whether this is all subject to re-interpretation based on the relatively recent description of Pa.
2. L58. The authors indicate that soft rot in konjac in their region is caused by Pa and this was shown by MLST typing, but references for this validation need to be provided. Again, it will not be clear to a reader whether the LAMP test developed here will be applicable in another area, where soft rot might be caused by another taxon (or taxa). How often is Pa the sole cause of soft rot in another region?
3. Table 1 and throughout the entire text. When accessing the CP001657.1 genome entry, locus accession PC1_1704 is an amidase of ~275 amino acids (~825 bp) with a signal sequence. This disagrees with the information presented (1809 bp, no signal, glycosyl transferase). When using blastn with primer F3, the target gene appears to be PC1_1695 at around nt 1932595. The authors should double check all entries in Table 1 and correct the text throughout.
4. Table 3. Are all the Pa isolates of one ST in the MLST typing? i.e is there any proof that any of YCC1-14 are different, NS1-5, WF1-5 etc? It could be that if one clone is represented in each series, only three strains are actually being texted.
5. Table 3 the entry for Pcc is unclear as there are four “-“ symbols that appear on only 2 of the three columns.
6. L184. In this reviewer’s copy, none of the PCR series lanes 6-10 appears positive. The authors should provide a better image and identify which lane is positive (as reported in L184).
Specific Points (language related)
1. Title: Since there are other taxa that can cause soft rot, and the LAMP has only been shown to be useful for one of these, the title should be less broad. I recommend replacing “soft-rot bacteria” (which implies more than one taxon) with “Pectobacterium aroidearum”.
2. L72 insert space after “date,”
3. L82 of the LAMP method
4. L84 to form a white precipitate of…
5. L95 of the LAMP assay..
6. L97 genes in the P…….
7. L102 translated L104 were selected for designing …
8. L139 after the (insert space)
9. L167 ensure that it is Figure 3 and not 43
10. L210 uncovering
11. L227 Therefore, the LAMP assay is
12. LL259 were then picked and suspended …. L262 until use as a
13. L265, L285 please replace Pcc with correct taxon
14. L272 Please provide references (plural) for these studies. If only a single study, a single reference should be inserted.
15. L326 16S
Author Response
Reviewer #2
Comment 1: The manuscript by Sun and colleagues is a revised manuscript (based on the multiple corrections) that describes the development of a sensitive and specific LAMP test for Pectobacterium aroidearum, a cause of soft rot in the konjac plant. Overall the manuscript is clearly written but there are apparent errors in locus ID (for the critical target identified and exploited) and the relationship of Pa versus Pcc needs to be better explained—especially since the manuscript has instances where Pa is being described as Pcc (mainly in the methods).
Response: Please see the responses below to Comments 2, 3 and 4 which provide details.
Specific Points (Scientific)
Comment 2: Throughout the ms (mostly), the Pcc language has been replaced by P. aroidearum. As originally written, the manuscript appears to have developed this assay for Pcc, but this has now been revised. Although there appears to be considerable misidentification in the Pcc group, the authors need to provide further explanation about whether bona fide Pcc isolates cause soft rot in konjac (as in ref 7) or whether this is all subject to re-interpretation based on the relatively recent description of Pa.
Response: Based on first round reviewer comments, we incorporated information from Nahban et al. (2013) which necessitated some Pcc being changed to P. aroidearum. We did further lab work to confirm that our isolates were P. aroidearum based on multilocus typing (8 genes as described in Nahban et al. 2013 and presented in Figure S1). We also obtained an actual Pcc isolate from the China General Microbiological Culture Collection Center, CGMCC, and tested the LAMP assay on it as well as other P. aroidearum-specific primers and found Pcc to be negative in these tests. We obtained diseased Konjac isolates from locations in Hubei province over 240 km apart and obtained 24 P. aroidearum isolates which we confirmed as pathogenic on konjac. The reference cited by the reviewer (Wu et al. 2011) only found Pcc since P. aroidearum had not been split off from it at that time (by Nahban et al. 2013). In addition, we checked the source of all Pcc that are presented in our dendrogram, and none of them were found to have originated from konjac. We also inoculated a Pcc strain (CGMCC1.141) from Chinese cabbage onto konjac corms, but no symptom was observed after 24 h or even 48 h.
To address the reviewer's concerns, we had added the following information to the manuscript:
"[Materials and Methods] 4.1 Isolation and pathogenicity of konjac soft rot pathogen
From August 2017 to September 2018, konjac soft rot samples were collected from Yichang and Enshi, the main konjac production areas in western Hubei province. Potential pathogens were isolated and purified by growth on LB solid medium. After incubation at 28°C for 24 h, single colonies were suspended in 4 ml LB liquid medium and incubated at 28°C for 12 h. Then 20 μL of this 1×108 CFU/mL suspension were placed onto local healthy konjac corms, and symptoms were evaluated 24 and 48 h later. For strains that caused soft rot of konjac corms in our pathogenicity test, 16S rDNA sequencing and MLST analysis were used to identify these strains.
[Results] 2.1 Isolation and pathogenicity of konjac soft rot pathogen
We obtained 115 isolations from konjac soft rot samples. There were 35 strains sharing a morphotype that we assumed was Pectobacterium, 15 strains resembling Dickeya, and the other 65 strains assumed to be Bacillus and other bacteria based on gross morphological features. All 115 strains were inoculated into konjac corms. The results showed that 50 strains caused rot 24 h after inoculation, and the normally white tissue turned yellowish brown or grayish white, showing a sticky paste texture and a distasteful odor, which was consistent with the field symptoms. After morphological observation, the 50 strains were divided into two types, one of which had 35 strains with colonies on LB solid medium that were milky white, nearly round, with smooth surface and neat edges, and slightly raised in the center of the colonies. The other 15 strains had colonies on LB solid medium that were milky white and wavy edges. We first used 16S rDNA sequencing and the 35 strains were placed as Pectobacterium while the 15 strains were placed as Dickeya. MLST analysis was then done to reveal that the 35 strains were P. aroidearum (Figure S1). Thus, P. aroidearum is the main soft rot pathogen of konjac in western Hubei, province, accounting for 70%, while Dickeya accounts for 30%. Therefore, the following experiments used P. aroidearum as the subject for further study.”
Comment 3: L58. The authors indicate that soft rot in konjac in their region is caused by Pa and this was shown by MLST typing, but references for this validation need to be provided. Again, it will not be clear to a reader whether the LAMP test developed here will be applicable in another area, where soft rot might be caused by another taxon (or taxa). How often is Pa the sole cause of soft rot in another region?
Response: From Figure S1, the 23 isolates of Pcc came from various countries and various hosts, but none were konjac. In this revision, we tested the pathogenicity of our Pcc strain (CGMCC1.141) from Chinese cabbage on konjac and found it to be non-pathogenic. The following has been added to the text. Line 113: "We tested the pathogenicity of Pcc strain (CGMCC1.141) on konjac, and found that it was unable to cause disease on konjac".
There is relative little research on konjac soft rot, and the older literature attributes the disease to Pcc before P. aroidearum was identified. Xu et al. (2011) found soft rot konjac from Shanxi province which is over 500 km away from our study sites, and their isolate M8 was found by Nahban et al. (2013) to be P. aroidearum. This has also been added to the revised manuscript (Page 2 Line 58): “And previous study found that strain M8, isolated from soft rot of konjac samples in Shaanxi province, is clustered together with P. aroidearum strain PC1 using phylogenetic analysis [10].’’ We cannot be absolutely certain that P. aroidearum is the major pathogen causing soft rot pathogen on konjac outside Hubei province, but there is no evidence that the taxon currently accepted as Pcc does cause soft rot of konjac.
Comment 4: Table 1 and throughout the entire text. When accessing the CP001657.1 genome entry, locus accession PC1_1704 is an amidase of ~275 amino acids (~825 bp) with a signal sequence. This disagrees with the information presented (1809 bp, no signal, glycosyl transferase). When using blastn with primer F3, the target gene appears to be PC1_1695 at around nt 1932595. The authors should double check all entries in Table 1 and correct the text throughout.
Response: Because we numbered the genes when we used the comparative genomics approach. 1704 was the number that we gave to the gene rather than the actual gene number from the database, which lead to misidentification. After carefully searching the original genome sequence again, the target gene should be PC1_1675, and the gene length is 1809 bp, at the location 1931694-1333502 bp. Gene ID information has all been re-verified and corrected throughout the manuscript.
Comment 5: Table 3. Are all the Pa isolates of one ST in the MLST typing? i.e is there any proof that any of YCC1-14 are different, NS1-5, WF1-5 etc? It could be that if one clone is represented in each series, only three strains are actually being texted.
Response: YCC1 to 14 were isolated from different konjac corms in Zhijiang and Changyang counties of Yichang (over 75 km distant), Hubei province. NS1 to 5 was isolated from different konjac corms in Enshi, Hubei province (furthest distance 240 km apart). WF1 to 5 were isolated from different konjac corms and stems in Wufeng county of Yichang, Hubei province (furthest distance 160 km apart). We did not test within county, but YCC1 and WF-2 showed no nucleotide differences in 4150 bp of the eight different genes, and they originated 170 km apart. There is a possibility that all tested P. aroidearum isolates are clonal, but further research is required on genetic diversity within this species.
Comment 6: Table 3 the entry for Pcc is unclear as there are four “-” symbols that appear on only 2 of the three columns.
Response: This is an issue when looking at the track-changed version since the third column was actually deleted in that version. There remained only two columns and the two "-" symbols did not fall squarely into those two column
Comment 7: L184. In this reviewer's copy, none of the PCR series lanes 6-10 appears positive. The authors should provide a better image and identify which lane is positive (as reported in L184).
Response: We have changed the image since Lane 6 actually shows weak band. We mention (Lines 193-197) that lanes 1 to 6 show a positive response.
Specific Points (language related)
Comment 8: Title: Since there are other taxa that can cause soft rot, and the LAMP has only been shown to be useful for one of these, the title should be less broad. I recommend replacing “soft-rot bacteria” (which implies more than one taxon) with “Pectobacterium aroidearum”.
Response: We have revised the title as: "Development of a Loop-Mediated Isothermal Amplification Method by Comparative Genomics for Rapid Detection of Pectobacterium aroidearum Soft Rot of Amorphophallus konjac"
Comment 9: L72 insert space after “date,”
Response: Corrected as suggested.
Comment 10: L82 of the LAMP method
Response: Corrected as suggested.
Comment 11: L84 to form a white precipitate of&
Response: Corrected as suggested.
Comment 12: L95 of the LAMP assay.
Response: Corrected as suggested.
Comment 13: L97 genes in the P&&.
Response: Corrected as suggested.
Comment 14: L102 translated
Response: Corrected as suggested.
Comment 15: L104 were selected for designing &
Response: We choose to retain "could be selected", since this part is regarding the 30 candidates, among which we only actually selected 9 for testing. Changing to "were selected" would make this incorrect.
Comment 16: L139 after the (insert space)
Response: Corrected as suggested
Comment 17: L167 ensure that it is Figure 3 and not 43
Response: Corrected as suggested.
Comment 18: L210 uncovering
Response: Corrected as suggested.
Comment 19: L227 Therefore, the LAMP assay is
Response: Corrected as suggested.
Comment 20: LL259 were then picked and suspended & L262 until use as a
Response: Corrected as suggested.
Comment 21: L265, L285 please replace Pcc with correct taxon
Response: Corrected as suggested
Comment 22: L272 Please provide references (plural) for these studies. If only a single study, a single reference should be inserted.
Response: We have revised this to state, "the genome is publicly available", since no publication has yet resulted from this work.
Comment 23: L326 16S
Response: Corrected as suggested.
Round 2
Reviewer 1 Report
The authors have addressed all of the raised issues in detail. Thank you for the opportunity for reviewing this important work.
Author Response
Thank you for your positive comment.
Reviewer 2 Report
The revised manuscript addresses most of the concerns raised in the previous review.
A few minor issues
1. It is a little confusing that the 1704-1 primer set is named after an irrelevant ORF. It would be better to refer to it as the 1675-1 set.
2. L32 infected
3. L52 the Enterobacteriaceae
4. L56/57 requires language revision
5. L58 that the main pathogen
6. L101 that was assumed to be Pecto……other 65 strains were assumed to be Bacillus…
7. L105 add “(data not shown)”
8. L123 remains awkwardly worded as written. Perhaps use “and these were thus suitable for …”
9. The fact that only one Pcc isolate was tested and none of the 15 soft-rot Dickeya isolates were tested, is a weakness. The fact that up to 30% of soft-rot in Konjac is caused by Dickeya, I would have thought these isolates would be important to check in a specificity assay. Especially as all symptomatic and several asymptomatic samples were positive in the assay.
10. L234 This is overstated as written since it cannot detect the 30% Dickeya mediated cases. It should be written that it can detect Pa mediated soft rot.
11. L298 Agro tumefaciens is a plant pathogen, so should be listed in the sentence above.
Author Response
Comment 1: It is a little confusing that the 1704-1 primer set is named after an irrelevant ORF. It would be better to refer to it as the 1675-1 set.
Response: As suggested, the primer set name has been changed to 1675-1.
Comment 2: L32 infected
Response: Corrected as suggested.
Comment 3: L52 the Enterobacteriaceae
Response: Corrected as suggested.
Comment 4: L56/57 requires language revision. “And previous study found that strain M8 was isolated from soft rot of konjac samples in Shaanxi province, which is clustered together with P. aroidearum strain PC1 using phylogenetic analysis [10].”
Response: reworded as: “Bacterial strain M8, isolated from konjac soft rot samples in Shaanxi province, was clustered together with P. aroidearum strain PC1 using phylogenetic analysis”
Comment 5: L58 that the main pathogen
Response: Corrected as suggested.
Comment 6: L101 that was assumed to be Pecto……other 65 strains were assumed to be Bacillus…
Response: Corrected as suggested.
Comment 7: L105 add “(data not shown)”
Response: Corrected as suggested.
Comment 8: L123 remains awkwardly worded as written. Perhaps use “and these were thus suitable for …”
Response: Corrected as suggested.
Comment 9: The fact that only one Pcc isolate was tested and none of the 15 soft-rot Dickeya isolates were tested, is a weakness. The fact that up to 30% of soft-rot in Konjac is caused by Dickeya, I would have thought these isolates would be important to check in a specificity assay. Especially as all symptomatic and several asymptomatic samples were positive in the assay.
Response: In the manuscript, several Dickeya strains (D. fangzhongdai, D. chrysanthemi and D. zeae) had been used in specificity assay. And among these, the pathogenic strain D. fangzhongdai YCH1 was isolated from konjac.
Comment 10: L234 This is overstated as written since it cannot detect the 30% Dickeya mediated cases. It should be written that it can detect Pa mediated soft rot.
Response: As suggested, the sentence has been revised as “Thus, this is the first description of specific detection of a konjac soft rot pathogen, and we developed a LAMP detection method specific for P. aroidearum which can detect P. aroidearum infections of konjac at an early stage of the disease
Comment 11: L298 Agro tumefaciens is a plant pathogen, so should be listed in the sentence above.
Response: Corrected as suggested.
Round 3
Reviewer 2 Report
Most of the comments (except #9) were addressed.
Author Response
Reviewer 2
Comment: The fact that only one Pcc isolate was tested and none of the 15 soft-rot Dickeya isolates were tested, is a weakness. The fact that up to 30% of soft-rot in Konjac is caused by Dickeya, I would have thought these isolates would be important to check in a specificity assay. Especially as all symptomatic and several asymptomatic samples were positive in the assay.
Response: As suggested, the 15 soft-rot Dickeya isolates have been tested in a specificity assay (new data). Negative reactions were found for all 15 Dickeya isolates using LAMP primer set 1675-1 or PCR primers 1675F3/B3. Please see Table 3 and revised Results. For Pcc, we were only able to obtain a single confirmed isolate (CGMCC1.141) from China General Microbiological Culture Collection Center, Beijing, China), and so this one was used in the specificity assay.
This manuscript is a resubmission of an earlier submission. The following is a list of the peer review reports and author responses from that submission.
Round 1
Reviewer 1 Report
This is a very well-written, and well-organized report of archival value. The methods are standard, the work is competent, and the findings are supported, but not surprising, and the work makes a contribution to the field.
Author Response
Response: Thank you for your positive comment.
Reviewer 2 Report
The goal of this work is useful and I learned useful information about konjac from this manuscript. The paper is appropriate for publication in this journal, once one major issue is remedied.
I have one significant concern about this paper. I think that they may have misidentified the species that they are working with. I think that this one major issue with this paper can be remedied mainly with a small amount of bioinformatics/phylogenetics computer analysis, and that this is required prior to publication.
Pc1, which was used for their initial primer design, was first reported as P. carotovorum, prior to several manuscripts that described multiple new Pectobacterium species. I believe that this strain has since been described as P. aroidearum. The authors could use a simple MLST analysis with type strains and Pc1 to confirm its identity. They should do this prior to publication.
I used several genome sequences for strains of which I am confident of the strain identity and the primers anneal to P. aroidearum and not to P. carotovorum. For this paper, the authors should include a BLAST analysis with type strains for Pectobacterium species to confirm whether their primers anneal to genome sequences from each type strain. This should also clarify which species they are working with.
Pectobacterium aroidearum is a species commonly found on monocot plants and this paper will be more interesting if this is the species they are working with since this species has only been reported from a few ornamental plants in Israel, as far as I know. Showing that it is causing disease on an important agricultural crop is interesting and also developing a test specifically for this species is novel.
Line 41: Is the authority for the scientific name of the plant and bacterial pathogen required by this journal?
Line 41: I could not figure out how glucomannan is used in oil. Is this for food products or fuel or a different use?
Line 41: Rewrite this sentence to: Konjac corm is rich in the hemicellulose glucomannan, which is widely used in many fields, including food additives, chemical engineering, medicine, and oil. Konjac is also an important source of starch [2,3].
Line 45: The authors write that bacterial soft rot causes a 30-50% reduction in Konjac yield. This is statement unclear – is it a 30-50% yield reduction across all producers? Does this yield reduction occur only in fields with the pathogen present? Or is this number for each infected plant? Please clarify this number.
Line 47: Consider removing passive voice throughout the manuscript since this can make the text easier to follow. For example:
Bacterial soft rot of konjac is mainly caused by Pectobacterium carotovorum subsp. carotovorum.
Line 51: describe what the timely control measures for this disease in konjac include. Is it rouging? Field destruction? Or other methods?
Line 84:There are many examples of misidentification in the Pectobacterium and Dickeya literature. Pc1 CP001657.1 is actually Pectobacterium aroidearum and not P. carotovorum. Since the methods section is at the end, it would help the reader if you specified on this line where Pc1 came from.
Table 1 – The Pc1 predicted proteins have additional information besides “hypothetical protein” that could be included in table 1. The protein family that they are part of is more useful than the GC%.
Table 3 – Why include results from the 27F-1492R PCR in table 2? This doesn’t provide any useful information.
Results from published PCR primers, such as the multiplex assay described by Potrykus et al. would be much more useful here since it would provide a direct comparison with a previously published assay for Pectobacterium detection. I suspect that the assay reported here may be superior – but the direct comparison would show this. (Potrykus M, Sledz W, Golanowska A, Slawiak M, Binek A, Motyka A, Zoledowska S, Czajkowski R, Lojkowska E. 2014. Simultaneous detection of major blackleg and soft rot bacterial pathogens in potato by multiplex polymerase chain reaction. Annals of Applied Biology 165:474-487.)
I don’t think that Fig. 2 is needed for this publication. The information could easily be provided in a table. Fig 3 shows how they are interpreting their LAMP results and this one is sufficient.
Line 133: change this to: Therefore, the sensitivity of LAMP detection was 100-fold greater than conventional PCR.
Note that a hyphen should be used between 100 and fold and that fold rather than folds should be used. It should be specified that the LAMP has greater sensitivity in this sentence.
Line 142: delete the phrase “the results showed” and instead write: The LAMP assay could detect Pcc at a concentration…
Section 2.5 –
In this section, it would be useful is a second assay were used to confirm the results from the field samples. For example, can the authors culture the bacteria from these samples to confirm that their assay is not giving false positives? Can they use a different PCR test (maybe one from Potrykus? I don’t know if these will work with their strains) to show that Pectobacterium is present? The scientific literature is filled with assays that give false positives with field samples, so providing confirmation with isolation or a second assay of some sort would strengthen this paper.
Author Response
Comment 1: The goal of this work is useful and I learned useful information about konjac from this manuscript. The paper is appropriate for publication in this journal, once one major issue is remedied. I have one significant concern about this paper. I think that they may have misidentified the species that they are working with. I think that this one major issue with this paper can be remedied mainly with a small amount of bioinformatics/phylogenetics computer analysis, and that this is required prior to publication. Pc1, which was used for their initial primer design, was first reported as P. carotovorum, prior to several manuscripts that described multiple new Pectobacterium species. I believe that this strain has since been described as P. aroidearum. The authors could use a simple MLST analysis with type strains and Pc1 to confirm its identity. They should do this prior to publication. I used several genome sequences for strains of which I am confident of the strain identity and the primers anneal to P. aroidearum and not to P. carotovorum. For this paper, the authors should include a BLAST analysis with type strains for Pectobacterium species to confirm whether their primers anneal to genome sequences from each type strain. This should also clarify which species they are working with. Pectobacterium aroidearum is a species commonly found on monocot plants and this paper will be more interesting if this is the species they are working with since this species has only been reported from a few ornamental plants in Israel, as far as I know. Showing that it is causing disease on an important agricultural crop is interesting and also developing a test
specifically for this species is novel.
Response: As suggested, we used MLST to analyze our strains (WF-2 and YCC1), P. aroidearum strain Pc1 and many other Pectobacterium spp. strains. The results showed our strains (WF-2 and YCC1) and Pc1 were definitely identified as P. aroidearum, and not P. carotovorum subsp. carotovorum. Hence, we now add the MLST analysis as Figure 1S and also reworded inappropriate descriptions thorough the manuscript. Please see MLST analysis below (Figure 1S).
Comment 2: Line 41: Is the authority for the scientific name of the plant and bacterial pathogen required by this journal?
Response: Yes, Latin authorities are used in this journal and we have included them at first mention of each species.
Comment 3: Line 41: I could not figure out how glucomannan is used in oil. Is this for food products or fuel or a different use? Line 41: Rewrite this sentence to: Konjac corm is rich in the hemicellulose glucomannan, which is widely used in many fields,4 including food additives, chemical engineering, medicine, and oil. Konjac is also an important source of starch [2,3,4].
Response: As reported previously, glucomannan could be used in oil industry. Usually, it is used as a weak gel, a corrosion inhibitor, an adhesion promoter or a plugging agent in the oil recovery process. We has also cited a new reference “Richard, 1991” in the manuscript. As suggested, we have rewritten this sentence as “Konjac corm is rich in the hemicellulose glucomannan, which is widely used in many fields, including food additives, chemical engineering, medicine and the oil recovery process. Konjac is also an important source of starch”.
Comment 4: Line 45: The authors write that bacterial soft rot causes a 30-50% reduction in Konjac yield. This is statement unclear – is it a 30-50% yield reduction across all producers? Does this yield reduction occur only in fields with the pathogen present? Or is this number for each infected plant? Please clarify this number.
Response: To make this statement clearer, we have reworded as follows: “In western Hubei province, as the most serious disease of konjac, bacterial soft rot generally causes a 30%–50% reduction in the corm yield of continuous cropping fields, and the loss of production even exceeds 80% in some planting areas.”
Comment 5: Line 47: Consider removing passive voice throughout the manuscript since this can make the text easier to follow. For example: Bacterial soft rot of konjac is mainly caused by Pectobacterium carotovorum subsp. carotovorum.
Response: As suggested, passive voice of some sentences has been removed to make
the text easier to follow.
Line 48, reworded as “Bacterial soft rot of konjac is mainly caused by Pectobacterium
carotovorum subsp. carotovorum”.
Line 89, reworded as “Among these, 30 genes had no matches in the NR protein
database, and these could be selected for designing and testing of specific primers.”
Line 102, reworded as “a total of six primer sets were specific to P. aroidearum by
conventional PCR. ”
Comment 6: Line 51: describe what the timely control measures for this disease in konjac include. Is it rouging? Field destruction? Or other methods?
Response: As suggested, we have added information about the control measures for this disease. Line 51, added as “Once the konjac corms are infected, there is no effective method to control bacterial soft rot. A common practice is to use fungicide-treated, healthy corms before planting”.
Comment 7: Line 84: There are many examples of misidentification in the Pectobacterium and Dickeya literature. Pc1 CP001657.1 is actually Pectobacterium aroidearum and not P. carotovorum. Since the methods section is at the end, it would help the reader if you specified on this line where Pc1 came from.
Response: We have re-identified the konjac soft-rot bacteria and Pc1 by MLST analysis as suggested and found these isolates are all actually P. aroidearum and not P. carotovorum. Please see response for Comment 1. The description for Pc1 has also been added into 2.1 section as “The genome of P. aroidearum strain PC1 (GenBank accession CP001657.1) was used for mining specific genes.”
Comment 8: Table 1 – The Pc1 predicted proteins have additional information besides “hypothetical protein” that could be included in table 1. The protein family that they are part of is more useful than the GC%.
Response: Since most of the target proteins are predicted as hypothetical proteins, alternatively, we use Signal P and SMART to analyze signal peptide and predict domains of these proteins, respectively. For analysis details, please see Table 1. Comment 9: Table 3 – Why include results from the 27F-1492R PCR in table 2? This doesn’t provide any useful information. Results from published PCR primers, such as the multiplex assay described by Potrykus et al. would be much more useful here since it would provide a direct comparison with a previously published assay for Pectobacterium detection. I suspect that the assay reported here may be superior – but the direct comparison would show this.
Response: Actually, the role of 27F-1492R PCR just confirmed DNA quality of each sample to avoid negative amplification due to low DNA quality of the sample. As suggested, we have deleted results from 27F-1492R PCR in Table 2. Comment 10: I don’t think that Fig. 2 is needed for this publication. The information could easily be provided in a table. Fig 3 shows how they are interpreting their LAMP results and this one is sufficient.
Response: As suggested, we have removed Fig. 2 from the manuscript.
Comment 11: Line 133: change this to: Therefore, the sensitivity of LAMP detection
was 100-fold greater than conventional PCR.
Response: Corrected as suggested.
Comment 12: Line 142: delete the phrase “the results showed” and instead write: The
LAMP assay could detect Pcc at a concentration
Response: Corrected as suggested.
Comment 13: Section 2.5 – In this section, it would be useful is a second assay were used to confirm the results from the field samples. For example, can the authors culture the bacteria from these samples to confirm that their assay is not giving false positives? Can they use a different PCR test (maybe one from Potrykus? I don’t know if these will work with their strains) to show that Pectobacterium is present? The scientific literature is filled with assays that give false positives with field samples, so providing confirmation with isolation or a second assay of some sort would strengthen this paper.
Response: Since we have no fresh samples for isolation during this season, we attempted a second assay, multiplex PCR to confirm the results from the field samples. However, no specific primers have been developed for detection of P. aroidearum. So we will leave this work for a future publication.